# Solution Behavior in the Vicinity of Characteristic Envelopes for the Double Slip and Rotation Model

**Yao Wang [1,2], Sergei Alexandrov [3,4] and Elena Lyamina [5,6,*]**

[1] National Key Laboratory for Precision Hot Processing of Metals, Harbin Institute of Technology, Harbin 150001, China; yao_wang86@163.com
[2] School of Mechanical Engineering, Hebei University of Technology, Tianjin 300130, China
[3] Ishlinsky Institute for Problems in Mechanics RAS, Moscow 119526, Russia; sergei_alexandrov@spartak.ru
[4] Federal State Autonomous Educational Institution of Higher Education, South Ural State University (national research university), Chelyabinsk 454080, Russia
[5] Division of Computational Mathematics and Engineering, Institute for Computational Science, Ton Duc Thang University, Ho Chi Minh City 700000, Vietnam
[6] Faculty of Civil Engineering, Ton Duc Thang University, Ho Chi Minh City 700000, Vietnam
* Correspondence: lyaminaea@tdtu.edu.vn; Tel.: +84-2837755024

**Abstract:** The boundary conditions significantly affect solution behavior near rough interfaces. This paper presents general asymptotic analysis of solutions for the rigid plastic double slip and rotation model in the vicinity of an envelope of characteristics under plane strain and axially symmetric conditions. This model is used in the mechanics of granular materials. The analysis has important implications for solving boundary value problems because the envelope of characteristics is a natural boundary of the analytic solution. Moreover, an envelope of characteristics often coincides with frictional interfaces. In this case, the regime of sticking is not possible independently of the friction law chosen. It is shown that the solution is singular in the vicinity of envelopes. In particular, the profile of the velocity component tangential to the envelope is described by the sum of the constant and square root functions of the normal distance to the envelope in its vicinity. As a result, some components of the strain rate tensor approach infinity. This finding might help to develop an efficient numerical method for solving boundary value problems and provide the basis for the interpretation of some experimental results.

**Keywords:** pressure-dependent plasticity; double slip and rotation model; envelope of characteristics; singularity; asymptotic analysis

## 1. Introduction

Models of pressure-dependent plasticity based on the Mohr–Coulomb yield criterion are widely used for describing deformation of granular materials and soil [1,2]. Boundary conditions for the flow of granular materials are attracting considerable interest due to their effect on solution behavior [3–10]. In the case of hyperbolic models, the envelope of characteristics is a natural boundary of the analytic solution. The envelope of characteristics often coincides with frictional interfaces. A distinguished feature of this boundary condition is that the regime of sliding occurs independently of the friction law chosen. In the case of pressure-dependent plasticity, such solutions have been found in [11–13]. In these works, the double shearing model proposed in [14] has been adopted. The solutions given in [11–13] show that the velocity field is singular (some derivatives approach infinity in the vicinity of envelopes).

The present paper deals with the rigid plastic double slip and rotation model under plane strain and axial symmetry conditions [15]. The solutions found in [11,12] are also solutions for this model if

the intrinsic spin involved in the double slip and rotation model vanishes. Other singular solutions for the rigid plastic double slip and rotation model have been derived in [16,17].

General asymptotic analyses of the singular behavior of solutions help to develop efficient numerical methods for solving boundary value problems and provide the basis for the interpretation of some experimental results. In the case of rigid plastic solids, such analyses have been carried out in [18] for isotropic perfectly plastic material, [19] for anisotropic perfectly plastic material, and [20] for a model that is used in the mechanics of polymers. These findings show that the exact asymptotic representation of solutions near envelopes of characteristics is affected by the shape of the yield surface and other constitutive equations. This paper supplies the exact asymptotic representation of solutions near envelopes of characteristics for the double slip and rotation model.

The applied aspect of the present paper is twofold. Firstly, the behavior of soil–structure interfaces is very important for predicting the overall response of soil–structure systems. It is known that a very narrow layer of soil adjacent to the structure controls the shearing deformation near the interface. Several models have been proposed for describing the behavior of material within this layer. One group of models introduces special constitutive equations in the vicinity of soil–structure interfaces (for example, [21–23]). The other group treats the interface layer as a geometric surface subject to certain physical conditions (for example, [24–26]). The exact asymptotic representation of solutions found in the present paper can be used in conjunction with the approach adopted in models of the first group. Secondly, suitable interface elements should be used to accurately simulate the behavior of material near soil–structure interfaces via the finite element method [27,28]. The present paper demonstrates that the solution may be singular. Therefore, traditional linear approximations with finite elements, for example [28], are not capable of representing the real solution. The exact asymptotic representation of solutions found can be used in conjunction with the extended finite element method [29].

## 2. Basic Equations

The double slip and rotation model is a model of pressure-dependent plasticity [15]. The version of the model considered in the present paper is rigid/plastic (i.e., the elastic portion of strain is neglected). The constitutive equations of the model are a pressure-dependent yield criterion and a flow rule. The present paper deals with the Mohr–Coulomb yield criterion. The flow rule has been formulated in [15].

### 2.1. Plane Strain Deformation

Let $(x_1, x_2)$ be an orthogonal coordinate system in planes of flow. The $x_2-$ coordinate curves are straight (Figure 1). Then, it is possible to assume with no loss of generality that the scale factor of these coordinate curves is unity. The scale factor of the $x_1-$ coordinate lines may be represented as

$$H_1 = 1 + \frac{x_2}{R(x_1)}. \tag{1}$$

Here $R(x_1)$ is the radius of curvature of the $x_1-$coordinate curve determined by the equation $x_2 = 0$ (S–line in Figure 1). The equilibrium equations referred to the coordinate system chosen are [30]

$$\frac{\partial \sigma_{11}}{\partial x_1} + \left[1 + \frac{x_2}{R(x_1)}\right]\frac{\partial \sigma_{12}}{\partial x_2} + \frac{2\sigma_{12}}{R(x_1)} = 0, \quad \left[1 + \frac{x_2}{R(x_1)}\right]\frac{\partial \sigma_{22}}{\partial x_2} + \frac{\partial \sigma_{12}}{\partial x_1} + \frac{(\sigma_{22} - \sigma_{11})}{R(x_1)} = 0. \tag{2}$$

Here, Equation (1) has been used. Furthermore, $\sigma_{11}$, $\sigma_{22}$ and $\sigma_{12}$ are the stress components referred to the $(x_1, x_2)-$ coordinate system.

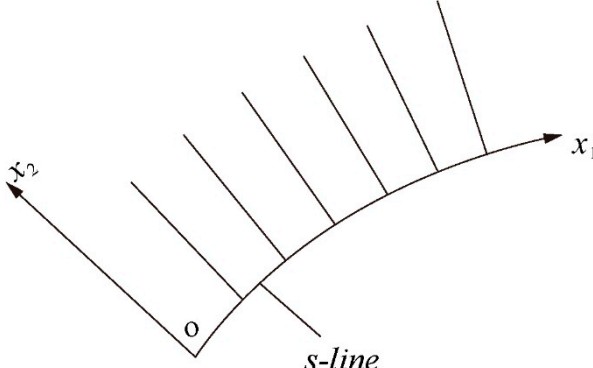

**Figure 1.** Orthogonal coordinate system $(x_1, x_2)$ near a given curve (S–line) under plane strain conditions.

As usual, it is convenient to introduce the following stress variables:

$$p = -\frac{1}{2}(\sigma_{11} + \sigma_{22}), \; q = \frac{1}{2}\sqrt{(\sigma_{11} - \sigma_{22})^2 + 4\sigma_{12}^2} > 0. \tag{3}$$

Then, the Mohr–Coulomb yield criterion reads

$$q - p\sin\phi = k\cos\phi. \tag{4}$$

Here $k$ is the cohesion and $\phi$ is the angle of internal friction. Both are material constants. Let $\psi$ be the orientation of the major principal stress $\sigma_1$ relative to the $x_1-$ direction (Figure 2). Then, using (3), the stress components referred to the $(x_1, x_2)-$ coordinate system are expressed as

$$\sigma_{11} = -p + q\cos 2\psi, \; \sigma_{22} = -p - q\cos 2\psi, \; \sigma_{12} = q\sin 2\psi. \tag{5}$$

Substituting (5) into (2) leads to

$$-\frac{\partial p}{\partial x_1} + \cos 2\psi \frac{\partial q}{\partial x_1} - 2q\sin 2\psi \frac{\partial \psi}{\partial x_1} + \left[1 + \frac{x_2}{R(x_1)}\right]\sin 2\psi \frac{\partial q}{\partial x_2} +$$
$$2q\cos 2\psi\left[1 + \frac{x_2}{R(x_1)}\right]\frac{\partial \psi}{\partial x_2} + \frac{2q\sin 2\psi}{R(x_1)} = 0,$$
$$-\left[1 + \frac{x_2}{R(x_1)}\right]\left(\frac{\partial p}{\partial x_2} + \cos 2\psi \frac{\partial q}{\partial x_2}\right) + 2q\left[1 + \frac{x_2}{R(x_1)}\right]\sin 2\psi \frac{\partial \psi}{\partial x_2} + \sin 2\psi \frac{\partial q}{\partial x_1} +$$
$$2q\cos 2\psi \frac{\partial \psi}{\partial x_1} - \frac{2q\cos 2\psi}{R(x_1)} = 0. \tag{6}$$

One can eliminate p in these equations using (4). As a result,

$$\left(\cos 2\psi - \frac{1}{\sin\phi}\right)\frac{\partial q}{\partial x_1} - 2q\sin 2\psi \frac{\partial \psi}{\partial x_1} + \left[1 + \frac{x_2}{R(x_1)}\right]\sin 2\psi \frac{\partial q}{\partial x_2} +$$
$$2q\cos 2\psi\left[1 + \frac{x_2}{R(x_1)}\right]\frac{\partial \psi}{\partial x_2} + \frac{2q\sin 2\psi}{R(x_1)} = 0,$$
$$-\left[1 + \frac{x_2}{R(x_1)}\right]\left(\cos 2\psi + \frac{1}{\sin\phi}\right)\frac{\partial q}{\partial x_2} + 2q\left[1 + \frac{x_2}{R(x_1)}\right]\sin 2\psi \frac{\partial \psi}{\partial x_2} + \sin 2\psi \frac{\partial q}{\partial x_1} +$$
$$2q\cos 2\psi \frac{\partial \psi}{\partial x_1} - \frac{2q\cos 2\psi}{R(x_1)} = 0. \tag{7}$$

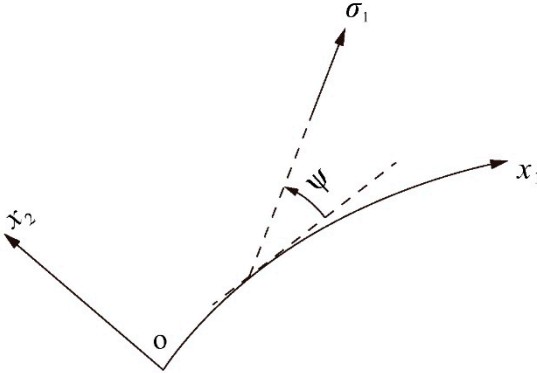

**Figure 2.** Orientation of the major principal stress.

The flow rule is [15]

$$\xi_{11} + \xi_{22} = 0, \ \xi_{12} \cos 2\psi - \frac{1}{2}(\xi_{11} - \xi_{22}) \sin 2\psi + (\omega_{12} + \Omega) \sin \phi = 0. \tag{8}$$

Here $\xi_{11}$, $\xi_{22}$ and $\xi_{12}$ are the strain rate components referred to the $(x_1, x_2)-$ coordinate system. Furthermore, $\omega_{12}$ is the only non-zero spin component and $\Omega$ is the intrinsic spin due to grain rotation. In general, the latter is an unknown variable which is governed by the equation of rotational motion. For the objective of the present paper it is sufficient to assume that $\Omega$ may be represented by a Taylor series in the vicinity of $x_2 = 0$. Using the first equation in (8) the second equation can be rewritten as

$$\xi_{12} \cos 2\psi - \xi_{11} \sin 2\psi + (\omega_{12} + \Omega) \sin \phi = 0. \tag{9}$$

The strain rate and spin components are expressed as

$$\xi_{11} = \left( \frac{\partial u_1}{\partial x_1} + \frac{u_2}{R(x_1)} \right) \left[ 1 + \frac{x_2}{R(x_1)} \right]^{-1}, \ \xi_{22} = \frac{\partial u_2}{\partial x_2},$$

$$2\xi_{12} = \frac{\partial u_1}{\partial x_2} + \left( \frac{\partial u_2}{\partial x_1} - \frac{u_1}{R(x_1)} \right) \left[ 1 + \frac{x_2}{R(x_1)} \right]^{-1}, \ 2\omega_{12} = \frac{\partial u_1}{\partial x_2} - \left( \frac{\partial u_2}{\partial x_1} - \frac{u_1}{R(x_1)} \right) \left[ 1 + \frac{x_2}{R(x_1)} \right]^{-1}. \tag{10}$$

Here $u_1$ and $u_2$ are the velocity components referred to the $(x_1, x_2)-$ coordinate system. Substituting (10) into (8) and (9) yields

$$\frac{\partial u_1}{\partial x_1} + \left[ 1 + \frac{x_2}{R(x_1)} \right] \frac{\partial u_2}{\partial x_2} = -\frac{u_2}{R(x_1)},$$

$$\left( \frac{\cos 2\psi}{\sin \phi} - 1 \right) \frac{\partial u_2}{\partial x_1} + \left( \frac{\cos 2\psi}{\sin \phi} + 1 \right) \left[ 1 + \frac{x_2}{R(x_1)} \right] \frac{\partial u_1}{\partial x_2} - \frac{2 \sin 2\psi}{\sin \phi} \frac{\partial u_1}{\partial x_1} = \tag{11}$$

$$\frac{2u_2 \sin 2\psi}{R(x_1) \sin \phi} - \frac{(\sin \phi - \cos 2\psi) u_1}{R(x_1) \sin \phi} - 2\Omega \left[ 1 + \frac{x_2}{R(x_1)} \right] = 0.$$

### 2.2. Axisymmetric Deformation

Let $(x_1, x_2)$ be an orthogonal coordinate system in the generic plane of a cylindrical coordinate system $(r, \theta, z)$ and $\gamma$ is the angle between the r-axis and the $x_1-$ direction measured from the r-axis anticlockwise (Figure 3). As in the case of plane strain deformation, the $x_2-$ lines are straight. Then, the scale factor of these coordinate lines are unity and the scale factor of the $x_1-$ coordinate lines is given by (1). The solution is independent of $\theta$. Moreover, the circumferential velocity and the shear stresses $\sigma_{1\theta}$ and $\sigma_{2\theta}$ vanish everywhere. The non-trivial equilibrium equations are [30]

$$\frac{\partial \sigma_{11}}{\partial x_1} + \left[1 + \frac{x_2}{R(x_1)}\right]\frac{\partial \sigma_{12}}{\partial x_2} + \frac{(\sigma_{11}-\sigma_{\theta\theta})}{r}\frac{\partial r}{\partial x_1} + \sigma_{12}\left[\left(1 + \frac{x_2}{R(x_1)}\right)\frac{\partial r}{r\partial x_2} + \frac{2}{R(x_1)}\right] = 0,$$

$$\frac{\partial \sigma_{22}}{\partial x_2} + \left[1 + \frac{x_2}{R(x_1)}\right]^{-1}\frac{\partial \sigma_{12}}{\partial x_1} + \frac{(\sigma_{22}-\sigma_{\theta\theta})}{r}\frac{\partial r}{\partial x_2} + \frac{(\sigma_{22}-\sigma_{11})}{R(x_1)+x_2} + \left(1 + \frac{x_2}{R(x_1)}\right)^{-1}\frac{\sigma_{12}}{r}\frac{\partial r}{\partial x_1} = 0. \tag{12}$$

It follows from the geometry of Figure 3 that $\partial r/\partial x_1 = (1 + x_2/R(x_1))\cos\gamma$ and $\partial r/\partial x_2 = -\sin\gamma$. Then, the equations in (12) become

$$\frac{\partial \sigma_{11}}{\partial x_1} + \left[1 + \frac{x_2}{R(x_1)}\right]\frac{\partial \sigma_{12}}{\partial x_2} + \frac{(\sigma_{11}-\sigma_{\theta\theta})}{r}\left[1 + \frac{x_2}{R(x_1)}\right]\cos\gamma + \sigma_{12}\left[\frac{2}{R(x_1)} - \left(1 + \frac{x_2}{R(x_1)}\right)\frac{\sin\gamma}{r}\right] = 0,$$

$$\frac{\partial \sigma_{22}}{\partial x_2} + \left[1 + \frac{x_2}{R(x_1)}\right]^{-1}\frac{\partial \sigma_{12}}{\partial x_1} - \frac{(\sigma_{22}-\sigma_{\theta\theta})\sin\gamma}{r} + \frac{(\sigma_{22}-\sigma_{11})}{R(x_1)+x_2} + \frac{\sigma_{12}\cos\gamma}{r} = 0. \tag{13}$$

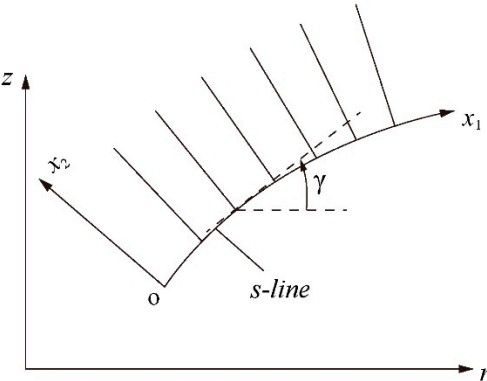

**Figure 3.** Orthogonal coordinate system $(x_1, x_2)$ near a given curve (S–line) under axially symmetric conditions.

There are several flow regimes associated with difference faces and edges of the Mohr–Coulomb yield criterion (Figure 4). One of the principal stresses is the circumferential stress $\sigma_{\theta\theta}$. The other principal stresses are denoted as $\sigma_1$ and $\sigma_2$. It is assumed that $\sigma_1 > \sigma_2$. In this case, there are two regimes in which the hypothesis of Haar and von Karmen, that $\sigma_1 = \sigma_{\theta\theta}$ or $\sigma_2 = \sigma_{\theta\theta}$, is satisfied. These regimes correspond to point A and F in Figure 4. Using (3) the yield criterion can be represented as

$$\sigma_{\theta\theta} = -p - mq, \ q = p\sin\phi + k\cos\phi \tag{14}$$

where $m = +1$ for regime A, and $m = -1$ for regime F. The equations in (5) are valid. Substituting (5) and the first equation in (14) into (13) yields

$$-\frac{\partial p}{\partial x_1} + \cos 2\psi \frac{\partial q}{\partial x_1} - 2q\sin 2\psi \frac{\partial \psi}{\partial x_1} + \left[1 + \frac{x_2}{R(x_1)}\right]\left(\sin 2\psi \frac{\partial q}{\partial x_2} + 2q\cos 2\psi \frac{\partial \psi}{\partial x_2}\right)+$$

$$\left[1 + \frac{x_2}{R(x_1)}\right]\frac{(\cos 2\psi+m)q\cos\gamma}{r} + \left[\frac{2}{R(x_1)} - \left(1 + \frac{x_2}{R(x_1)}\right)\frac{\sin\gamma}{r}\right]q\sin 2\psi = 0,$$

$$-\frac{\partial p}{\partial x_2} - \cos 2\psi \frac{\partial q}{\partial x_2} + 2q\sin 2\psi \frac{\partial \psi}{\partial x_2} + \left[1 + \frac{x_2}{R(x_1)}\right]^{-1}\left(\sin 2\psi \frac{\partial q}{\partial x_1} + 2q\cos 2\psi \frac{\partial \psi}{\partial x_1}\right)+$$

$$\frac{(\cos 2\psi-m)q\sin\gamma}{r} - \frac{2q\cos 2\psi}{R(x_1)+x_2} + \frac{q\sin 2\psi\cos\gamma}{r} = 0. \tag{15}$$

One can eliminate $p$ here using the second equation in (14). Then, the equations in (15) become

$$
\begin{aligned}
&\left(\cos 2\psi - \tfrac{1}{\sin\phi}\right)\frac{\partial q}{\partial x_1} - 2q\sin 2\psi \frac{\partial \psi}{\partial x_1} + \left[1 + \tfrac{x_2}{R(x_1)}\right]\left(\sin 2\psi \frac{\partial q}{\partial x_2} + 2q\cos 2\psi \frac{\partial \psi}{\partial x_2}\right) + \\
&\left[1 + \tfrac{x_2}{R(x_1)}\right]\frac{(\cos 2\psi + m)q\cos\gamma}{r} + \left[\tfrac{2}{R(x_1)} - \left(1 + \tfrac{x_2}{R(x_1)}\right)\tfrac{\sin\gamma}{r}\right]q\sin 2\psi = 0, \\
&-\left(\cos 2\psi + \tfrac{1}{\sin\phi}\right)\frac{\partial q}{\partial x_2} + 2q\sin 2\psi \frac{\partial \psi}{\partial x_2} + \left[1 + \tfrac{x_2}{R(x_1)}\right]^{-1}\left(\sin 2\psi \frac{\partial q}{\partial x_1} + 2q\cos 2\psi \frac{\partial \psi}{\partial x_1}\right) + \\
&\frac{(\cos 2\psi - m)q\sin\gamma}{r} - \frac{2q\cos 2\psi}{R(x_1) + x_2} + \frac{q\sin 2\psi \cos\gamma}{r} = 0.
\end{aligned}
\tag{16}
$$

The flow rule is

$$
\xi_{11} + \xi_{22} + \xi_{\theta\theta} = 0, \quad \xi_{12}\cos 2\psi - \frac{1}{2}(\xi_{11} - \xi_{22})\sin 2\psi + (\omega_{12} + \Omega)\sin\phi = 0.
\tag{17}
$$

Here $\xi_{\theta\theta}$ is the circumferential strain rate. The equations in (10) are valid. Moreover, $\xi_{\theta\theta} = u_r/r$ where $u_r$ is the radial velocity. It follows from the geometry of Figure 3 that $u_r = u_1\cos\gamma - u_2\sin\gamma$. Using this equation and (10) it is possible to transform (17) into

$$
\begin{aligned}
&\frac{\partial u_1}{\partial x_1} + \left[1 + \tfrac{x_2}{R(x_1)}\right]\frac{\partial u_2}{\partial x_2} = \frac{(u_2\sin\gamma - u_1\cos\gamma)}{r}\left[1 + \tfrac{x_2}{R(x_1)}\right] - \frac{u_2}{R(x_1)}, \\
&\left(\tfrac{\cos 2\psi}{\sin\phi} - 1\right)\frac{\partial u_2}{\partial x_1} + \left(\tfrac{\cos 2\psi}{\sin\phi} + 1\right)\left[1 + \tfrac{x_2}{R(x_1)}\right]\frac{\partial u_1}{\partial x_2} - \frac{\sin 2\psi}{\sin\phi}\frac{\partial u_1}{\partial x_1} + \\
&\left[1 + \tfrac{x_2}{R(x_1)}\right]\frac{\sin 2\psi}{\sin\phi}\frac{\partial u_2}{\partial x_2} = \frac{(\cos 2\psi - \sin\phi)u_1}{R(x_1)\sin\phi} + \frac{\sin 2\psi u_2}{R(x_1)\sin\phi} - 2\Omega\left[1 + \tfrac{x_2}{R(x_1)}\right].
\end{aligned}
\tag{18}
$$

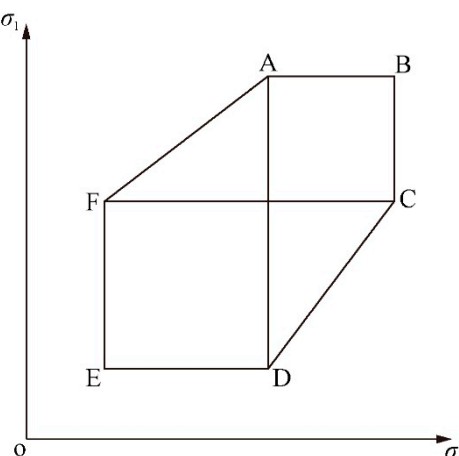

**Figure 4.** Mohr–Coulomb yield locus under axially symmetric conditions.

## 3. Characteristic Analysis

### 3.1. Plane Strain Deformation

Multiplying the first equation in (7) by $\cos 2\psi$, the second by $\sin 2\psi$ and summing gives

$$
\left(1 - \frac{\cos 2\psi}{\sin\phi}\right)\frac{\partial q}{\partial x_1} - \frac{\sin 2\psi}{\sin\phi}\left[1 + \frac{x_2}{R(x_1)}\right]\frac{\partial q}{\partial x_2} + 2q\left[1 + \frac{x_2}{R(x_1)}\right]\frac{\partial \psi}{\partial x_2} = 0.
\tag{19}
$$

Multiplying the first equation in (7) by $\sin 2\psi$, the second by $-\cos 2\psi$ and summing gives

$$
\frac{\sin 2\psi}{\sin\phi}\frac{\partial q}{\partial x_1} - \left(1 + \frac{\cos 2\psi}{\sin\phi}\right)\left[1 + \frac{x_2}{R(x_1)}\right]\frac{\partial q}{\partial x_2} + 2q\frac{\partial \psi}{\partial x_1} = \frac{2q}{R(x_1)}.
\tag{20}
$$

Assume that S – curve in Figure 1 is a characteristic curve. Then, $\psi = \pm(\pi/4 + \phi/2)$ on this curve [14]. In particular, $\partial\psi/\partial x_1 = 0$. Equation (20) supplies the characteristic relation in the form

$$\pm \cot\phi \frac{\partial q}{\partial x_1} - \frac{2q}{R(x_1)} = 0. \tag{21}$$

In the proceeding, it is assumed that S – curve in Figure 1 is an envelope of characteristics. Therefore, Equation (21) is not satisfied even though $\psi = \pm(\pi/4 + \phi/2)$ on this curve. It is possible only if

$$\left| \frac{\partial q}{\partial x_2} \right| \to \infty \tag{22}$$

as $x_2 \to 0$. In this case, the second term in (20) reduces to the expression $0 \cdot \infty$ at $x_2 = 0$.

### 3.2. Axisymmetric Deformation

Multiplying the first equation in (16) by $\cos\psi$, the second by $\sin\psi$ and summing gives

$$\left(1 - \frac{\cos 2\psi}{\sin\phi}\right)\frac{\partial q}{\partial x_1} - \frac{\sin 2\psi}{\sin\phi}\left[1 + \frac{x_2}{R(x_1)}\right]\frac{\partial q}{\partial x_2} + 2q\left[1 + \frac{x_2}{R(x_1)}\right]\frac{\partial \psi}{\partial x_2} = F_1. \tag{23}$$

Multiplying the first equation in (16) by $\sin\psi$, the second by $-\cos\psi$ and summing gives

$$\frac{\sin 2\psi}{\sin\phi}\frac{\partial q}{\partial x_1} - \left(1 + \frac{\cos 2\psi}{\sin\phi}\right)\left[1 + \frac{x_2}{R(x_1)}\right]\frac{\partial q}{\partial x_2} + 2q\frac{\partial \psi}{\partial x_1} = F_2. \tag{24}$$

The terms $F_1$ and $F_2$ include no derivatives.

Assume that S – curve in Figure 3 is a characteristic curve. Then, $\psi = \pm(\pi/4 + \phi/2)$ on this curve [31]. In particular, $\partial\psi/\partial x_1 = 0$. Equation (24) supplies the characteristic relation in the form

$$\pm \cot\phi \frac{\partial q}{\partial x_1} = F_2. \tag{25}$$

In the proceeding, it is assumed that S – curve in Figure 3 is an envelope of characteristics. Therefore, Equation (25) is not satisfied even though $\psi = \pm(\pi/4 + \phi/2)$ on this curve. It is possible only if

$$\left| \frac{\partial q}{\partial x_2} \right| \to \infty \tag{26}$$

as $x_2 \to 0$. In this case, the second term in (24) reduces to the expression $0 \cdot \infty$ at $x_2 = 0$.

## 4. Asymptotic Analysis

The asymptotic analysis below is based on the assumptions that all stress and velocity components are bounded everywhere and all derivatives with respect to $x_1$ are bounded everywhere, and the solution is represented by power series in $x_2$ near the curve $x_2 = 0$ (Figures 1 and 3).

### 4.1. Plane Strain Deformation

Let $\psi_f$ be the value of $\psi$ at $x_2 = 0$. For definiteness, it is assumed that $\psi_f = \pi/4 + \phi/2$. The case $\psi_f = -(\pi/4 + \phi/2)$ can be treated in a similar manner. The condition (22) requires that

$$q = q_0 + q_1 x_2^\alpha + o\left(x_2^\alpha\right) \tag{27}$$

as $x_2 \to 0$ and

$$0 < \alpha < 1. \tag{28}$$

Here $q_0$ and $q_1$ are independent of $x_2$. The first term in (19) is bounded. Therefore, Equations (27) and (19) combine to give

$$\psi = \psi_f + \psi_1 x_2^\alpha + o(x_2^\alpha) \tag{29}$$

as $x_2 \to 0$. Turning to Equation (20), the first and third terms of this equation are bounded. The term on its right-hand side is also bounded. Therefore, the second term must be bounded as $x_2 \to 0$. Expanding $\cos 2\psi$ in the Taylor series in the vicinity of $\psi = \psi_f$ one gets

$$1 + \frac{\cos 2\psi}{\sin \phi} = -2 \cot \phi (\psi - \psi_f) + o[(\psi - \psi_f)] \tag{30}$$

as $\psi \to \psi_f$. Using (27), (29) and (30) the second term in (20) is represented as

$$\left(1 + \frac{\cos 2\psi}{\sin \phi}\right)\left[1 + \frac{x_2}{R(x_1)}\right]\frac{\partial q}{\partial x_2} = -2\alpha q_1 \psi_1 \cot \phi x_2^{2\alpha-1} + o(x_2^{2\alpha-1}) \tag{31}$$

as $x_2 \to 0$. Substituting (27), (29) and (31) into (20) shows that $x_2^{2\alpha-1} = O(x_2^\alpha)$ or $x_2^{2\alpha-1} = O(1)$ as $x_2 \to 0$. The former yields $\alpha = 1$, which contradicts (28). The latter results in

$$\alpha = \frac{1}{2}. \tag{32}$$

Then, (27) and (29) become

$$q = q_0 + q_1 \sqrt{x_2} + o(\sqrt{x_2}) \text{ and } \psi = \psi_f + \psi_1 \sqrt{x_2} + o(\sqrt{x_2}) \tag{33}$$

as $x_2 \to 0$, respectively. The distribution of $p$ neat the curve is readily found from (4) and (33) as

$$p = \frac{q_0 - k \cos \phi}{\sin \phi} + \frac{q_1}{\sin \phi} \sqrt{x_2} + o(\sqrt{x_2}) \tag{34}$$

as $x_2 \to 0$.

It is now necessary to demonstrate that (33) is compatible with the plastic flow rule. The velocity components are represented as

$$u_1 = U_0 + U_1 x_2^\beta + o(x_2^\beta) \text{ and } u_2 = V_0 + V_1 x_2^\lambda + o(x_2^\lambda) \tag{35}$$

as $x_2 \to 0$. Here $U_0$, $U_1$, $V_0$ and $V_1$ are independent of $x_2$. Moreover, $\beta > 0$ and $\lambda > 0$. The first equation in (11) shows that $\partial u_2/\partial x_2 = O(1)$ as $x_2 \to 0$. Therefore $\lambda = 1$, and the second equation in (34) becomes

$$u_2 = V_0 + V_1 x_2 + o(x_2) \tag{36}$$

as $x_2 \to 0$. Substituting (30), (33), (35) and (36) into the second equation in (11) results in

$$2\psi_1 V_1 \beta \cot \phi \sqrt{x_2} x_2^{\beta-1} = O(x_2) \tag{37}$$

as $x_2 \to 0$. It follows from this equation that $\beta = 1/2$ and the first equation in (35) becomes

$$u_1 = U_0 + U_1 \sqrt{x_2} + o(\sqrt{x_2}) \tag{38}$$

as $x_2 \to 0$.

To the best of the authors' knowledge, two semi-analytical solutions for the double slip and rotation are available [16,17]. The asymptotic analysis of these solutions shows that they satisfy (33), (34), (36) and (38). Several semi-analytic solutions for the double-shearing model, which is

another widely used model of pressure-dependent plasticity, have been derived in [11–13] and [16]. The asymptotic analysis of these solutions also shows that they satisfy (33), (34), (36), and (38).

*4.2. Axisymmetric Deformation.*

It is seen from (19), (20), (23) and (24) that the left-hand sides of the equations under plane strain and axial symmetry coincide. The right-hand sides of these equations have no effect on asymptotic analysis. Therefore, the equations in (33) and (34) are valid for axisymmetric deformation. The circumferential stress is readily determined from the first equation in (14).

Turning to the velocity equations in (18), the derivative $\partial u_2 / \partial x_2$ in the second equation can be eliminated using the first equation. As a result,

$$
\begin{aligned}
&\frac{\partial u_1}{\partial x_1} + \left[ 1 + \frac{x_2}{R(x_1)} \right] \frac{\partial u_2}{\partial x_2} = \frac{(u_2 \sin \gamma - u_1 \cos \gamma)}{r} \left[ 1 + \frac{x_2}{R(x_1)} \right] - \frac{u_2}{R(x_1)}, \\
&\left( \frac{\cos 2\psi}{\sin \phi} - 1 \right) \frac{\partial u_2}{\partial x_1} + \left( \frac{\cos 2\psi}{\sin \phi} + 1 \right) \left[ 1 + \frac{x_2}{R(x_1)} \right] \frac{\partial u_1}{\partial x_2} - \frac{2 \sin 2\psi}{\sin \phi} \frac{\partial u_1}{\partial x_1} + \\
&= \left[ 1 + \frac{x_2}{R(x_1)} \right] \frac{(u_1 \cos \gamma - u_2 \sin \gamma) \sin 2\psi}{r \sin \phi} + \frac{(\cos 2\psi - \sin \phi) u_1}{R(x_1) \sin \phi} + \frac{2 u_2 \sin 2\psi}{R(x_1) \sin \phi} - 2\Omega \left[ 1 + \frac{x_2}{R(x_1)} \right].
\end{aligned}
\tag{39}
$$

The left-hand sides of these equations coincide with the corresponding left-hand sides of the equations in (11). Therefore, Equations (36) and (38) are also valid.

## 5. Conclusions

The exact asymptotic representation of solutions in the vicinity of envelopes of characteristics has been found for the double slip and rotation model under plane strain and axially symmetric conditions. This representation coincides with the behavior of specific solutions found in [11–13,16,17]. This representation also coincides with that for isotropic rigid plastic solids obeying an arbitrary smooth yield criterion [18], and for some anisotropic yield criteria [19]. It is seen from Equations (33), (34) and (38) that the stress and velocity fields are singular in the vicinity of the envelope of characteristics, in the sense that some derivatives of the stress and velocity components approach infinity near the envelope.

The applied aspect of the asymptotic representation of solutions derived in the present paper is twofold. Firstly, as known from other models of rigid plasticity, numerical solutions using traditional finite elements do not converge if the exact solution is described by equations similar to (33), (34) and (38) [32,33]. The asymptotic representation given in (33), (34) and (38) can be used in conjunction with the extended finite element method, outlined in [29], for developing an accurate numerical approach for solving corresponding boundary value problems.

Secondly, the asymptotic representation given in (33), (34) and (38) can be used in conjunction with models for soil–structure interfaces [21–23]. In particular, an envelope of characteristics usually coincides with the interface between a structure and material, for example between a structure and soil. In this case, experiment confirms some mathematical features of the general theoretical solution. If an envelope of characteristics coincides with the interface then the regime of sliding occurs independently of the friction law chosen. This means that the friction stress is independent of the roughness of the contact surface. It is in general known that the surface roughness affects the material behavior near the interface [34,35]. However, this effect is usually significant only if the surface roughness is small enough. For example, in the case of the interface between Yongdinghe sand and low carbon steel the friction stress is practically independent of the surface roughness if $R_n > R_{cr} = 0.1$ where $R_n$ is the relative interface roughness and $R_{cr}$ is its critical value [35]. In [35], the soil–structure interfaces are classified as smooth or rough depending on the value of $R_n$. When $R_n < R_{cr}$ the interface is smooth. If $R_n > R_{cr}$ then the interface is rough. In the latter case, strain localization occurs near the interface. This feature of material behavior is in agreement with (38) as this equation predicts that the shear strain rate approaches infinity as $x_2 \to 0$. Moreover, it has been found in [35] that the residual friction angle is close to the internal friction angle of sand if $R_n > R_{cr}$. This feature is in agreement with the

second equation in (33). Therefore, it is reasonable to assume that the condition that an envelope of characteristics coincides with the soil–structure interface is adequate when $R_n > R_{cr}$. In this case, it is natural to expect the similarity between the behavior of soil and interface. This similarity has been investigated in [23].

**Author Contributions:** Conceptualization, S.A.; writing, Y.W.; formal analysis, E.L. All authors have read and agreed to the published version of the manuscript.

**Funding:** This research was funded by Natural Science Foundation of Hebei Province of China with Grant No. E2019202224 and the RFBR (Project No.18-51-76001).

**Conflicts of Interest:** The authors declare no conflict of interest.

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
