# Peer review of "Solution Behavior in the Vicinity of Characteristic Envelopes for the Double Slip and Rotation Model"

_applsci, doi:10.3390/app10093220_

Round 1
Reviewer 1 Report
The reviewer is sorry to have to declare that he probably made a mistake when accepting to review this manuscript. Reading the title and browsing through the abstract had not allowed him to realise beforehand that the subject of the manuscript was about the mechanics of granular material, a domain in which the reviewer has no competence (the reviewer was only intrigued about the meaning of “an envelope of characteristics”, and maybe a little curious to discover about this concept). At least this means that the authors have to modify the title, or at least the abstract, in such a way as to announce clearly what is the domain of the work. The present title and abstract would be OK for a journal exclusively devoted to the mechanics of granular materials, which is certainly not the case for Applied Sciences.
Anyway, the reviewer has made the effort to read the manuscript to the end. He has found it formally very well written, with excellent English language, very good argumentation, and very extensive reference to the state of the art in literature. Although he cannot judge whether or not the mathematics is completely sound, the reviewer has the feeling that the manuscript is of good quality, and likely to be acceptable as such. Nevertheless, the Editor should try to find another reviewer specialist in the mathematical modelling of the deformation of granular media.
Author Response
We received the following message from the editor:
"The first reviewer basically excused himself/herself as a
non-expert, so his/her review doesn't count."
Nevertheless, we took your comment into account.
Reviewer 2 Report
The present manuscript is a slight modification of the published paper
[A] Solution Behavior near Envelopes of Characteristics for Certain Constitutive Equations Used in the Mechanics of Polymers, 2019, Materials 12(17):2725, DOI: 10.3390/ma12172725
by the same authors. This paper [A] is not cited in the manuscript.
Moreover, I find a discrepancy between equations (2) and the standard equations in polar coordinates in the case x1 = θ, x2 = r – r0 for a fixed positive r0.
Therefore, the manuscript should be rejected.
Author Response
Please see the file attached.

Reviewer 3 Report
The authors present rigid-plastic (Mohr-Coulomb) slip-rotation asymptotic solutions at a frictional boundary and detect singularities. The solutions are currently represented as a set of equations without proper analysis of singularities and comparison with computational results. This obscures the meaning.
(1) The authors should demonstrate their claim that "This representation coincides with the behavior of specific solutions found in [11 -13, 16, 17].", by illustrating graphically the features that correspond to solutions in [11 -13, 16, 17].
(2) The claim: "The asymptotic representation given in (33), (34) and (38) can be used in conjunction with the extended finite element method [28] for developing an accurate numerical approach for solving corresponding boundary value problems." is dubious. It brings up the question of elasticity. What would addition of elasticity do to the observed singularities? Are the singularities artifacts of rigid-plastic assumption? Why would anyone use finite elements to solve a rigid-plastic problem if addition of elasticity makes the problem numerically tractable? A simple finite element run of elastic-plastic material would illustrate what happens with singularities if elasticity is included. elastic-plastic Mohr-Coulomb materials are available in most commercial FE packages.
Author Response
Please see the file attached

Round 2
Reviewer 2 Report
The authors give the required answers. I suppose that the similarity of mathematical models has to be noted despite the difference of the considered physical processes. Hence, the reference [20] should be cited.
The paper can be published in “Applied Sciences”.
Author Response
[20] has been included in the list of references
Reviewer 3 Report
The authors have chosen to misunderstand my comments from the first review and to ignore them. They are in fact refusing to put their very specialized small-niche contribution in a general context, or to in any way evaluate the relevance of their result. I don't believe that these results are publishable in the present form.
Author Response
Please find the file attached.
